# From Desire to Dread—A Neurocircuitry Based Model for Food Avoidance in Anorexia Nervosa

**DOI:** 10.3390/jcm10112228

**Published:** 2021-05-21

**Authors:** Guido K. W. Frank

**Affiliations:** 1Department of Psychiatry, University of California San Diego, San Diego, CA 92093, USA; gfrank@health.ucsd.edu; Tel.: +1-858-246-5073; 2Eating Disorders Center for Treatment and Research, UC San Diego Health, San Diego, CA 92121, USA

**Keywords:** anorexia nervosa, dopamine, reward, fear, conditioning, cortex, striatum, nucleus accumbens, hypothalamus

## Abstract

Anorexia nervosa is a severe psychiatric illness associated with food avoidance. Animal models from Berridge et al. over the past decade showed that environmental ambience, pleasant or fear inducing, can trigger either appetitive (desire) or avoidance (dread) behaviors in animals via frontal cortex, nucleus accumbens dopamine D1 and D2 receptors, and hypothalamus. Those mechanisms could be relevant for understanding anorexia nervosa. However, models that translate animal research to explain the psychopathology of anorexia nervosa are sparse. This article reviews animal and human research to find evidence for whether this model can explain food avoidance behaviors in anorexia nervosa. Research on anorexia nervosa suggests fear conditioning to food, activation of the corticostriatal brain circuitry, sensitization of ventral striatal dopamine response, and alterations in hypothalamic function. The results support the applicability of the animal neurocircuitry derived model and provide directions to further study the pathophysiology that underlies anorexia nervosa.

## 1. Introduction

Anorexia nervosa (AN) is a severe psychiatric illness associated with food restriction, severe emaciation, intense fear of gaining weight or becoming fat, and a perception of being overweight despite severe underweight [1]. Developing models for AN’s underlying etiology and pathophysiology has been difficult due to a complex interplay between neurobiological, psychological and environmental factors [2]. This has also limited the development of more effective treatments, including medication interventions [3,4].

Altered reward or salient stimulus processing has been found in AN in studies from many laboratories and using different methodologies [5,6,7]. There is evidence that reward circuit alterations are related to neuroendocrine abnormalities that occur during starvation, while altered reward reactivity may also have genetic underpinnings in AN. However, cause and effect of abnormal salient stimulus processing and weight loss in AN have not been resolved [8,9,10].

Brain reward response is important for learning and behavior selection, and involves frontal cortical brain regions, as well as subcortical regions such as the ventral striatum [11]. Reward-based learning and associated behavior selection can be goal directed, habitual, or a combination of those, and these concepts have been used to integrate neurobiology and behavior in AN to mechanistically explain food avoidance. One model here is the habit-based model, which implicates frontostriatal circuits producing automatic instrumental stimulus-response behaviors (habits) to drive persistent food avoidance [12]. One recent report in support of that model indicated that food restriction was strongly related to habit strength, but not cognitive restraint, using subscales derived from the Eating Pathology Symptoms Inventory [13]. Habit-based model-free learning does not rely on conceptualization of outside events, yet learning is based on the “utility of outcomes encountered on past interactions with the environment” [14]. Others have described the ego syntonic nature of food restriction in AN, based on altered value processing of food stimuli, high anxiety and altered fear extinction, suggesting that AN is rather motivated by goal-directed instrumental behaviors, and thus a conscious choice [15,16]. This model-based learning drives goal-directed instrumental behaviors based on “internal maps of the environment and cognitive styles” [14]. One study also suggested that there may be subgroups of individuals with AN, one whose illness may rather be driven by goal-directed choice and another with a more habit-driven pathology [17]. Studies from our lab have applied a Pavlovian reward learning model studying neuronal response to unexpected receipt or omission of salient stimuli, which is based on dopamine related reward or salient stimulus processing [18,19,20]. This led to the development of a model where anxiety driven motivation to restrict food becomes extreme, alters biological functions across brain and body, including a sensitization of the dopamine circuitry, and perpetuates a self-reinforcing cycle of fear, food restriction and weight loss [21]. The Pavlovian model encompasses both model-free (dopamine signal-based associative learning) and model-based (a neutral stimulus becomes conditioned based on motivational salience of an unconditioned stimulus) aspects [14]. A benefit of applying the Pavlovian model in AN research is that it has a strong neurotransmitter-related underlying framework, which can be used to study specific dopamine-related pathophysiology. While the above models provide insight into brain circuitry that may be specific to AN’s pathophysiology, we are still searching for a more in-depth understanding of how neurotransmitter-based circuitry drives food avoidance.

Over the past decade, animal studies have developed a significant body of research that can be used to build upon our understanding of the underlying neurobiology of AN. Specifically, those models can help tease apart neurotransmitter systems involved in food avoidance, as well as clarify the contributions of learning and conditioning. A recent line of research reviewed by Castro, Cole, and Berridge has shown that the nucleus accumbens, a key structure for dopaminergic and opioidergic function in the reward cycle, contains both areas to drive eating (desire) but also fearful reactions (dread) [22]. Whether the desire or dread response is activated is dependent on whether the individual is in a self-perceived safe environment or an environment that is experienced as anxiety provoking or hostile. Frontal cortical connections relay this information to the subcortical circuitry, involving gamma-aminobutyric acid (GABA) and glutamate to drive or inhibit regional and receptor-specific dopamine circuit response [23]. Importantly, dopamine D1 receptor stimulation in the rostral nucleus accumbens can generate appetitive food intake, while simultaneous dopamine D1 and D2 receptor stimulation in the caudal nucleus accumbens initiates fearful defensive behaviors. This model has not been studied in humans or AN specifically. Furthermore, dopaminergic projections from the nucleus accumbens to the hypothalamus, a brain region that integrates bodily signals such as hunger, regulate food intake, via a fear-mediated circuitry that enables the organism to override appetitive drive in the presence of perceived danger, which could have implications for AN [21,24].

This review surveyed the literature over the past decade to gather support for the applicability of Castro et al.’s model to AN. We hypothesized that we would find evidence for a model that integrates conditioned fear-driven, top-down control over food intake, subcortical dopamine-driven associative learning, and hypothalamic food intake control, to explain why food restriction-sensitized dopaminergic circuits do not drive food intake, but rather defensive behaviors in AN.

## 2. Materials and Methods

State-of-the-art review of the literature on both animal and human studies over the past ten years to describe the current state of knowledge and priorities for future investigation and research [25].

We searched pubmed (https://pubmed.ncbi.nlm.nih.gov) on 5 March 2021, using the search terms, anorexia nervosa + reward (327 records), anorexia nervosa + hypothalamus (123 records), anorexia nervosa + food control circuitry (15 records), anorexia nervosa + dopamine (74 records), anorexia nervosa + glutamate (22 records), and anorexia nervosa + GABA (21 records).

After initial review of all abstracts, 222 papers were downloaded and further reviewed. To reduce bias or effects of small sample sizes in human brain imaging studies, we limited the results to studies with at least 20 subjects per group [26].

A total of 109 articles were identified as relevant to the topic as they related to the model. Thirty-one duplicates were removed, resulting in 78 articles reviewed here. Other articles were excluded due to small sample size, not related to the topic or being a review article.

## 3. Results

### 3.1. Corticostriatal Circuits and Food Choice

Richard and Berridge previously described that activation or inhibition of frontal cortical regions can activate a nucleus accumbens response that is associated with either desire or dread response [27]. Specifically, orbitofrontal cortex stimulation increased the desire to eat via reducing caudal nucleus accumbens activity, while infralimbic cortex (human ventral, subgenual anterior cingulate) stimulation suppressed both eating and fearful behavior via nucleus accumbens response. The latter was conceptualized that anterior cingulate activation may attenuate generation of any intense motivations. Dopamine signaling may trigger nucleus accumbens response for desire or dread, mediated via direct and indirect corticostriatal GABAergic and glutamatergic connections [22,28].

Fear and anxiety are elevated in individuals with AN, especially when presented with food [29], which activate the biological stress circuitry and alter reward system response [30]. Food choice in AN is highly determined by fear of weight gain and the desire to avoid calories, which was associated with elevated dorsal striatal brain response [31,32]. Furthermore, striatal-prefrontal cortex functional connectivity predicted inversely high versus low caloric food intake. Studies in AN after recovery showed that in a fasted state and in response to repeated predictable tasting of sucrose solution or receipt of monetary reward, the ventral striatum was less activated compared to controls, but related to harm avoidance, a measure for trait anxiety [33,34]. Those studies emphasized that a hungry state does not activate reward circuits in AN as in controls, but brain response may be rather mediated by negative emotionality. One study using monetary reward showed that the gut hormone ghrelin may impact reward-related decision making in AN, implicating body homeostasis and feeding state in this process [35]. A caveat is that while the association between dopamine reward circuitry and ghrelin is appealing, evidence for its involvement in human psychopathology is otherwise largely lacking [36,37]. A longitudinal study in AN suggested that excessive self-control over food intake may diminish with treatment, as indicated by stronger default mode network activation during a reward task after partial weight restoration [38]. A few studies investigated the association between reward value and brain response in AN. While one study found elevated anterior cingulate activation in AN compared to controls using a taste task, others did not find differences in neural reward value response after recovery [39,40], suggesting state-dependent changes.

In summary, recent behavioral and brain circuit research in AN indicates and confirms that because of the presence of high anxiety when presented with food, illness-driven food choices are associated with elevated dorsal striatal response, and striatal-prefrontal cortex connectivity and huger hunger do not recruit reward circuits in AN as they do in controls. Whether food-value computation is altered in AN requires further study, but a difficulty may be the distinction of positive or negative value versus motivational salience and stimulus strength as the main driver of brain response. Those studies are in line with Castro et al.’s model, indicating that emotional state, such as fear of weight gain, drives defensive behavior response, or dread in the context of food presentation. This may lead to an activation of the frontostriatal brain circuitry, potentially contributing to habit formation [27]. Consistent with that notion is a recent study using an animal model for AN that found that suppression of corticostriatal connections improved cognitive flexibility and was associated with less weight loss, suggesting that reducing higher-order cortical input to the subcortical reward circuitry might be beneficial to treat AN, by reducing cognitive rigidity and anxiety [41].

### 3.2. Ventral Striatal Reward System and Food Intake Control

The neuroscience-based model that we explore indicates that the nucleus accumbens, part of the ventral striatum, codes both appetitive food approach as well as defensive fear behaviors, on a trajectory from rostral to caudal nucleus accumbens [42]. Specifically, glutamate antagonism or GABA A receptor agonistic agents, which modulate dopamine release in the rostral nucleus accumbens shell increase appetitive behaviors such as eating but lead to opposite behaviors such as conditioned place avoidance and defensive behaviors in the caudal accumbal region. Further studies then showed that dopamine D1 receptors are needed for stimulating eating in the rostral nucleus accumbens, but simultaneous dopamine D1 and D2 receptor signaling in the caudal nucleus accumbens is needed to drive fear response [43,44]. Altogether, the data suggest that corticolimbic glutamate and subcortical GABA signals can drive desire or dread via nucleus accumbens dopamine activation, depending on the accumbal subregion [45]. The dopamine system adapts to food restriction with enhanced neuronal response [46,47,48,49,50,51], and sensitized dopamine circuitry in AN could be a vulnerability for excessive desire or dread response to salient stimuli [9,21,52,53].

The activity-based anorexia (ABA) model is an animal model that uses rodents to model key symptoms of AN. When rodents have time-limited access to food while having free access to a running wheel, they become hyperactive and lose body weight excessively [54]. The ABA model identified changes in glutamate receptor expression that persisted after body weight recovery [55]. This could indicate that food restriction and weight loss disrupt glutamate signaling in the brain, which could alter dopamine function and subsequently appetitive behavior or dread response. Another rodent model indicated that the application of the dopamine precursor L-dopa normalized self-starvation in acetylcholine receptor knock-out mice, raising the question of whether altered acetylcholine expression could be part of AN pathophysiology and disrupt dopamine response [56]. Another recent study using the ABA model in mice showed that overexpression of the dopamine D2 receptor in the nucleus accumbens induced weight loss in females who had limited (7 h per day) access to food, with or without running wheel access [57]. That study pointed to two important hypotheses. First, that elevated dopamine receptor D2 activity is associated with starvation. This was further supported in another ABA mouse model that indicated that dopamine transmission may be indicative of vulnerability to develop AN [58]. Second, there are sex-specific relationships between dopamine transmission and weight loss. Other studies indicated that fear-cued food avoidance is prolonged in females versus males, which could be part of the mechanism for why females are predominantly affected by AN, as prolonged dread response and food avoidance may be necessary to set off the vicious cycle of weight loss [21,59,60]. A difficulty associated with the ABA model has been that cognitive-emotional aspects of anorexia nervosa such as fear of weight gain and body image distortion cannot easily be modeled. A few studies have found mechanisms that could be relevant for AN, such as agouti-related peptide effects on operant feeding conditioning or the interaction between anxiety and genotype that may drive food restriction [61,62]. Translation of those animal studies into human research could be fruitful to explore interactions between neurobiology and behavior.

Human studies that directly measure brain dopamine circuit markers are sparse. One study measuring dopamine D2 receptor distribution indicated no difference in AN compared to controls in striatal or other brain regions [63]. Our lab has taken a different approach, using a Pavlovian reward learning paradigm that elicits the dopamine-related prediction error response, although this is not a direct measure of dopamine activation. The difference between an expectation and outcome yields the so-called prediction error, a dopamine-associated signal that reinforces new associations [11,64]. The direction of the prediction error is indicated by its sign which indicates a better (positive) or worse (negative) outcome than expected. The absolute value of the prediction error, which we have been applying, reflects the degree of deviation of the outcome from the expectation and is related to surprise or conceptualized as a motivational salience signal rather than taking better or worse outcome into account [65,66]. Studies from our lab have repeatedly shown that prediction error response is elevated in AN compared to controls, suggesting elevated dopamine signaling [18,19,20,67]. We hypothesized that anxious conditioning to eating and the fear of weight gain trigger the nucleus accumbens-mediated dread response to food stimuli. This response is exaggerated due to the food restriction-facilitated sensitization and elevation of nucleus accumbens dopamine response, which enables the individual to override normal hunger cues. We found that prediction error brain response is related to dopamine receptor genotype in healthy controls, but conclusive data in AN are lacking and it is yet unclear whether there is genetic vulnerability in AN for elevated dopamine responsiveness [68,69].

In summary, the model implicates nucleus accumbens dopamine function in both appetitive food approach and dread or fearful response, depending on emotional state or environmental ambience. The reviewed animal and human data on AN brain neurobiology indicate elevated dopamine-related brain response, which could provide a vulnerability to enhanced fear conditioning and resulting in dread response in the context of negatively biased food stimuli [28]. This mechanism may drive food avoidance and weight loss and lead to insensitivity to hunger cues. Importantly, fear conditioning and dopamine response appear to be sex specific, which provides a framework for the female predominance of AN.

### 3.3. Hypothalamic Eating Regulation

The hypothalamus has long been known to integrate bodily signals including hunger to motivate food intake. More recently, a dopamine D1 receptor-mediated circuitry from the ventral striatum to the hypothalamus has been identified that is fear driven and able to override hunger signals [24]. Castro et al.’s model supports this circuit response, but also suggests that both dopamine D1 and D2 receptor expressing neurons project to the hypothalamus. Earlier studies have shown that normal endocrine function in response to weight loss is generally preserved in AN, yet, individuals with AN are able to override hunger signals generated by the hypothalamic endocrine response [70]. Those observations support the notion that higher-order fear-motivated executive function circuits control or suppress signals to stimulate eating. Several newer studies have investigated the hypothalamus’ role in AN pathophysiology and identified new hypothalamus-specific targets for further investigation.

Neuroscience research in animals has found various indices of hypothalamus involvement in AN. One study found altered hypothalamus endocannabinoid tone in the ABA model, which could alter dopamine tone and thus affect both feeding drive or fear response [71]. The ABA model also associated altered mitochondrial response, elevated corticotropin-releasing factor neuron expressions and activated autophagy in the hypothalamus with weight loss, indicating structural and functional changes in response to starvation [72,73]. Another study using the ABA model identified increases in GABA A receptor α4 and δ subunits in ABA mice in response to underweight, which may be related to increased anxiety during stress, driving hyperactivity and weight loss [74]. Expression of those GABA receptor subunits is dependent on steroid sex hormones, which could indicate a specific interaction between female sex and receptor expression mediated by starvation [75]. Those studies led to the hypothesis that hypothalamic GABAergic neurotransmission could determine resilience or vulnerability to developing AN [76].

Several studies indicated long lasting effects in the hypothalamus from starvation. Chronically underfed mice, when re-fed, showed overexpression of hypothalamic leptin receptor mRNA, while melanocortin receptors remained upregulated from the underweight state [77]. Using a low weight chick animal model for AN, studies showed decreased agouti-related peptide and increased melanocortin-3 receptor expression in the hypothalamus, and elevated anorexigenic melanocortin signaling was hypothesized to block the orexigenic, appetite effect of neuropeptide Y [78,79].

Human brain imaging studies found various structural or functional hypothalamic alterations in AN. AN was associated with higher structural connectivity between hypothalamus and orbitofrontal cortex and amygdala, which could contribute to altered functional activity [80,81]. Furthermore, effective connectivity during sucrose tasting was directed from the ventral striatum to the hypothalamus in adults and youth with AN, while the direction was the opposite in healthy controls [19,80]. A study that delivered glucose solution to the stomach via a tube to circumvent effects of sweet taste and thus associated cognitions and emotions showed reduced hypothalamic activation and functional connectivity with nucleus accumbens in AN compared to controls [82]. That study implicated abnormal function of the hypothalamus in response to body homeostatic signals in the underweight state in AN. Another indicator for altered hypothalamic function in AN was paradoxical decrease in glutamate response to feeding compared to controls, which may alter dopamine release [83].

Kisspeptin describes a family of peptide hormones associated with emotional and sexual function that was found to increase food intake via modulation of hypothalamic GABAergic signaling [84]. In AN, kisspeptin was negatively related to exercise, thus excessive physical activity in AN may downregulate this hormone and contribute to reducing hypothalamic food drive [85]. A study that investigated human derived stem cells showed differences in AN compared to controls in gene expression after fasting in the hypothalamus, although the function or behavioral effects of those genes requires clarification [86].

Taken together, the hypothalamus is a key structure to drive food intake, and recent studies have identified a specific dopamine D1 receptor-mediated circuitry to control hypothalamic appetitive drive via the ventral striatum. Human studies found structural and functional differences in the hypothalamus in AN compared to controls, which included higher structural connectivity, and altered dopamine, glutamate and melanocortin expression. Several studies stand out. Conscious sugar water delivery activated the ventral striatal-to-hypothalamic effective connectivity, opposite to controls, suggesting activation of a fear-driven circuitry to override hunger signals. Unconscious delivery of sugar solution was associated with lower hypothalamic activation, suggesting reduced hypothalamic baseline responsiveness, which could be a genetic predisposition or result from starvation.

## 4. Discussion

The model described by Castro et al. that is derived from animal studies describes a circuitry between frontal cortex, ventral striatum and hypothalamus that drives either food approach or fearful defensive behaviors. Depending on whether the animals were at ease in a familiar environment or were stressed, either rostral accumbal dopamine D1 receptors mediated desire and appetitive behaviors, or simultaneous caudal nucleus accumbens D1 and D2 receptors mediated dread, producing fearful defensive actions. Nucleus accumbens to hypothalamus connections further mediate food approach by allowing or interrupting hypothalamic signaling of caloric needs of the organism.

The here reviewed literature supports Castro et al.’s model applicability to support the following model for AN (Figure 1). Fear of weight gain leads to a negative ambience around food, as food and eating have become conditioned fear-inducing stimuli, leading to negative ruminations [87,88]. This negative emotional state, especially in the presence of food, then may activate caudal accumbal dopamine D1 and D2 receptors to drive a defensive fear response or dread to food. The ventral striatal-hypothalamus dopamine D1 receptor-mediated fear circuitry may then enable the individual with AN to override hunger signals. Fear conditioning, dread response, and food avoidance are all at least in part mediated via dopaminergic circuits. Those dopaminergic circuits are hypersensitized due to starvation, which enhances fear conditioning, dread, and food avoidance response.

This model allows to test specific hypotheses to improve AN pathophysiology and associated illness-driven behaviors. One direction is to alter negative ambience and negative bias toward food, which is part of cognitive behavioral psychotherapy. Whether behavioral interventions such as attention bias modification are successful in AN to reduce anxiety and improve food intake, remains to be seen and may be age dependent [89]. Behavioral research should specifically be directed toward fear extinction to food and eating.

A biological approach could apply in humans where “turning off” of frontal input was able to improve survival in the ABA model [41]. In fact, a recent pilot study in AN using repetitive transcranial magnetic stimulation found less over-control for food choices, suggesting more cognitive flexibility [90]. Thus, transcranial magnetic stimulation should be investigated systematically for its role in alleviating cognitive rigidity to alleviate AN related fears.

There is indication that individuals with AN are prone to fast fear-related learning in situations that involve potential punishment or negative outcome, while fear extinction might be impaired [91,92]. This is further supported by research that showed impairments in feedback learning in AN [93]. Whether elevated fear response in AN is related to heightened sensitivity to salient stimuli or poor sensory discrimination as proposed for other disorders requires further study [94,95,96,97].

Another biological research direction would be targeting the dopamine system and normalizing hyperresponsiveness to normalize dread response. Animal research that applied dopamine D2/3 receptor antagonists in the ABA model found an improved survival rate compared to controls. However, dopamine D2 receptor antagonists overall have not been effective in treating AN [98,99]. The dopamine D1 and D2 receptor antagonist olanzapine has shown some benefits for AN in an outpatient setting, and clinically it is frequently used to treat acute severe anxiety or agitation in AN [100]. It is possible that the combined D1 and D2 receptor blockage reduces fear and dread response, although it has limited effect on appetitive food response. While the evidence is still scant, there is now some emerging evidence that the partial D2 receptor agonist aripiprazole could be helpful in targeting illness related preoccupations and supporting insight in AN [101,102,103,104,105]. This may be consistent with animal studies that showed elevated survival of the ABA rodent model in response to stimulating dopamine neurons in the nucleus accumbens [106]. Furthermore, frontal cortex dopamine D2 receptor stimulation enhances cognitive flexibility and fear extinction, which could also account for aripiprazole’s effects to improve treatment outcome [107].

Lastly, there has been an increased focus on metabolic abnormalities in AN over the past decade, indicating that AN is associated with gene expression for lower body mass index [108,109]. Whether those genes are strictly related to metabolism or also to, for instance, reduced salience network activation in AN as a trait that reduces the innate drive to forage for food should be investigated further [110].

In summary, Berridge et al.’s studies, using animal research, can be aligned with human studies to support a working model to conceptualize a neurobiological framework in AN that involves the corticostriatal-hypothalamic brain circuitry. This model provides a background for testable hypotheses to study specific neurotransmitter and neurocircuit-based pathophysiology and treatment response in AN.

## Figures and Tables

**Figure 1 jcm-10-02228-f001:**
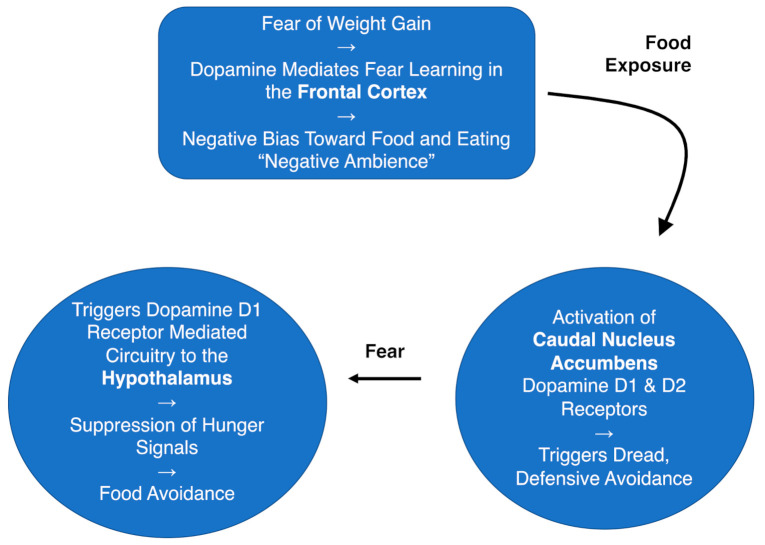
Hypothesized model for food avoidance in anorexia nervosa based on Castro et al., 2015 [22].

## Data Availability

Not applicable.

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
