# Peer review of "From Desire to Dread—A Neurocircuitry Based Model for Food Avoidance in Anorexia Nervosa"

_jcm, 2021, doi:10.3390/jcm10112228_

Round 1

Reviewer 1 Report

The paper is an interesting review of the existing literature about the neurobiology of anorexia nervosa. Papers about this topic are lacking in the field and this is an interesting summary that could be the base for future studies. I really appreciate the paper and I have only some minor comments for the author:

  • it could be helpful for the reader the inclusion of the date of the search of the papers that is also required by the PRISMA rules
  • I think the last phrase of the paper sounds out of context, I suggest to  evaluate a different conclusion of the paper 

I don't have any other comments for the author.

Author Response

The paper is an interesting review of the existing literature about the neurobiology of anorexia nervosa. Papers about this topic are lacking in the field and this is an interesting summary that could be the base for future studies. I really appreciate the paper and I have only some minor comments for the author:

Response: Thank you very much for the very positive feedback, it is very much appreciated.

  • it could be helpful for the reader the inclusion of the date of the search of the papers that is also required by the PRISMA rules

Response: The date of the search was March 5, 2021. This has been added to the manuscript.

  • I think the last phrase of the paper sounds out of context, I suggest to evaluate a different conclusion of the paper 

Response: Thank you for the suggestion. That sentence has been edited as follows: “In summary, Berridge et al.’s studies, using animal research, can be aligned with human studies to support a working model to conceptualize a neurobiological framework in AN that involves the cortico-striatal-hypothalamic brain circuitry. This model provides a background for testable hypotheses to study specific neurotransmitter and neurocircuit-based pathophysiology and treatment response in AN.”

Reviewer 2 Report

This paper proposes a model for understanding the pathophysiology of anorexia nervosa, and reviews some of the literature that serves to support the proposed model. The integration of this literature is a large challenge, and is impressive. The concerns about the paper come largely from the assertion that this is a systematic review, which it is not. In addition, some sections are not sufficiently precise.

A few important limitations undermine the value of this paper. The paper states that it is a systematic review of the literature from the past 10 years. Usually, when a time frame like that is selected, it is because there was a prior systematic review and the current is an update. That is not the case here, and by limiting the review to 10 years the findings (or the papers selected) are inherently biased. The author transparently states that the review was conducted to “gather support” and “find evidence.” This is antithetical to the concept of a systematic review. The author might do better to frame this as a “focused review” or, what it really is, which is the presentation of a potentially useful model of AN.

Throughout the paper, the author talks about “basic science” when he should be more precise that he is describing animal studies. One significant limitation here is that while animal models of overeating have proven useful as a basis for obesity research, animal studies of feeding behavior have not yet easily translated to findings in anorexia nervosa. The paper would be strengthened by discussing this issue, perhaps through more explicit description of the ABA model and the more recent animal models that show voluntary food avoidance (for example Zeltser research, and Andermann research).  

The Methods (and the PRISMA flow diagram) do not clarify why studies might have been excluded. Only articles deemed relevant to the model the authors propose were included. As stated above, this is not a systematic review. The paper should be revised without Methods and Results sections.

The section reviewing ventral striatal reward and food intake is quite interesting, and is an impressive review and integration of animal studies using the ABA model of AN. It needs to be clear that it is a review of animal research (not of “neuroscience” more broadly), and the language would benefit from being more precise (i.e., being careful about the use of emotion-language when the data are all behavioral in animals). It would also be useful to describe what the ABA model is, and what it does and does not represent about AN. Obviously, all animal models are limited. It would help this review to be clear about that.

The review of cognition and emotion research is considerably less clear than the reward section. It seems to be a quick review of some neuroimaging findings – not clear why only those were selected – and the review is so cursory as to be unhelpful to the reader. The summary of this section does not follow from the prior paragraphs.

The section on the hypothalamus needs to be more clear that the ABA model is limited in certain ways, since many assertions about AN are made based on animal findings. This section also seems to disregard all the early literature that indicated that hypothalamic functioning in AN is not disturbed – that is, the signals of starvation are present and neuroendocrine findings show the expected signs of starvation and signals to eat. Patients override these signals by not eating. The more recent findings are intriguing and allow for updating of consideration of the role of the hypothalamus, but by limiting the review the author has biased the findings. This may be an area where the author could contribute some testable hypotheses.

In the Discussion, would suggest changing much of the language to say “may” – as in, “the negative emotional state may activate caudal…” as the author has not conclusively demonstrated this. Furthermore, the hypotheses that the author proposes testing in order to evaluate the model are not articulated. In general, the Discussion needs editing for clarity of language.

Small note – the methods says “we searched” but there is only one author on the paper. This is disconcerting.

Small note – there are places where the references don’t seem to apply to the point being made.

Specific comments on the Intro:

The author states that “most deaths” from AN “occur between 16 and 29 years.” That does not seem to match the papers cited. One of them does indicate a higher SMR in that age range, but a different one indicates that deaths increase with duration of illness, and are higher in later decades. 

The Intro should be more clear that the research from Castro et al is referring to animal research. It would be useful to know how much of this work has been translated to human basic science research.

I found the first half of the Intro off-topic from what the paper actually is. Most of the paper is actually a review of findings from animal models, and these serve as the basis for the author’s proposed model, if I understood correctly. If so, Im not sure the description of habit formation is relevant. The author seems to be quickly describing several different models of learning and the result is not that clear. Since the habit formation literature isn’t really part of the science that gets reviewed, it may not be useful in the Intro.

Figure 1 is not really explained, and therefore doesn’t contribute to the paper.

Specific comments on the Abstract:

The abstract strings together many jargon terms, and this ultimately obscures the meaning. In addition, some terms are used loosely in ways that seem potentially inaccurate. For example, “food intake control circuitry” is not really established in humans; “central neural response” and “environmental ambience” are unclear phrases. It is not clear in the abstract what the “basic science model” is that the author is proposing.

Author Response

Thank you for the constructive criticism. The points raised have been addressed point by point below.

  1. This paper proposes a model for understanding the pathophysiology of anorexia nervosa, and reviews some of the literature that serves to support the proposed model. The integration of this literature is a large challenge, and is impressive. The concerns about the paper come largely from the assertion that this is a systematic review, which it is not. In addition, some sections are not sufficiently precise.

Response: I agree. It is by definition not a systematic review. The most appropriate description would be a State-of-the-Art review, the search strategy “aims for comprehensive searching of current literature”, and the analysis focuses on “current state of knowledge and priorities for future investigation and research” (Grant & Booth, 2009). We have edited the description of the review as such.

“This state-of-the-art review surveyed basic and human research of the past ten years… .”

“State-of-the-art review of the literature over the past ten years to describe the current state of knowledge and priorities for future investigation and research (Grant & Booth, 2009).

  1. A few important limitations undermine the value of this paper. The paper states that it is a systematic review of the literature from the past 10 years. Usually, when a time frame like that is selected, it is because there was a prior systematic review and the current is an update. That is not the case here, and by limiting the review to 10 years the findings (or the papers selected) are inherently biased. The author transparently states that the review was conducted to “gather support” and “find evidence.” This is antithetical to the concept of a systematic review. The author might do better to frame this as a “focused review” or, what it really is, which is the presentation of a potentially useful model of AN.

Response: Please see #1, I agree and have revised the description, accordingly, including removal of reference to the PRISMA guidelines. In contrast, I have added a description of the state-of-the-art review approach.

  1. Throughout the paper, the author talks about “basic science” when he should be more precise that he is describing animal studies. One significant limitation here is that while animal models of overeating have proven useful as a basis for obesity research, animal studies of feeding behavior have not yet easily translated to findings in anorexia nervosa. The paper would be strengthened by discussing this issue, perhaps through more explicit description of the ABA model and the more recent animal models that show voluntary food avoidance (for example Zeltser research, and Andermann research).  

Response: The term “basic” has been removed and replaced with terms that emphasize the animal research described.

The following description of the ABA model has been added: “The activity-based anorexia (ABA) model is an animal model that uses rodents to model key symptoms of AN. When rodents have time-limited access to food while having free access to a running wheel, they become hyperactive and lose body weight excessively (Chowdhury, Chen, & Aoki, 2015).”

The following text has also been added: “A difficulty associated with the ABA model has been that cognitive-emotional aspects of anorexia nervosa such as fear of weight gain and body image distortion cannot easily be modeled. A few studies have found mechanisms that could be relevant for AN, such as agouti-related peptide effects on operant feeding conditioning or the interaction between anxiety and genotype that may drive food restriction (Jikomes, Ramesh, Mandelblat-Cerf, & Andermann, 2016; Madra & Zeltser, 2016). Translation of those animal studies into human research could be fruitful to explore interactions between neurobiology and behavior.”  

  1. The Methods (and the PRISMA flow diagram) do not clarify why studies might have been excluded. Only articles deemed relevant to the model the authors propose were included. As stated above, this is not a systematic review. The paper should be revised without Methods and Results sections.

Response: The methods section has been revised and references to systematic review have been removed.

  1. The section reviewing ventral striatal reward and food intake is quite interesting, and is an impressive review and integration of animal studies using the ABA model of AN. It needs to be clear that it is a review of animal research (not of “neuroscience” more broadly), and the language would benefit from being more precise (i.e., being careful about the use of emotion-language when the data are all behavioral in animals). It would also be useful to describe what the ABA model is, and what it does and does not represent about AN. Obviously, all animal models are limited. It would help this review to be clear about that.

Response: I agree, please see above. Text has been added to address this concern – see #3.

  1. The review of cognition and emotion research is considerably less clear than the reward section. It seems to be a quick review of some neuroimaging findings – not clear why only those were selected – and the review is so cursory as to be unhelpful to the reader. The summary of this section does not follow from the prior paragraphs.

Response: Thank you for pointing this out. That was a misnomer to some degree. The focus is in fact on cortico-striatal circuits and food choice, and the title 3.1. is now reflecting this topic. The literature reviewed covers brain imaging studies on this area of research over the past decade.

  1. The section on the hypothalamus needs to be more clear that the ABA model is limited in certain ways, since many assertions about AN are made based on animal findings. This section also seems to disregard all the early literature that indicated that hypothalamic functioning in AN is not disturbed – that is, the signals of starvation are present and neuroendocrine findings show the expected signs of starvation and signals to eat. Patients override these signals by not eating. The more recent findings are intriguing and allow for updating of consideration of the role of the hypothalamus, but by limiting the review the author has biased the findings. This may be an area where the author could contribute some testable hypotheses.

Response: The text has been edited to address this concern and provide more clarification: “The hypothalamus has long been known to integrate bodily signals including hunger to motivate food intake. More recently, a dopamine D1 receptor mediated circuitry from the ventral striatum to the hypothalamus has been identified that is fear driven and able to override hunger signals (O'Connor et al., 2015). Castro et al.’s model supports this circuit response, but also suggests that both dopamine D1 and D2 receptor expressing neurons project to the hypothalamus. Earlier studies have shown that normal endocrine function in response to weight loss is generally preserved in AN, yet, individuals with AN are able to override hunger signals generated by the hypothalamic endocrine response (Monteleone, Castaldo, & Maj, 2008). Those observations support the notion that higher order fear-motivated executive function circuits control or suppress signals to stimulate eating. Several newer studies have investigated the hypothalamus’ role in AN pathophysiology and identified new hypothalamus specific targets for further investigation.”

  1. In the Discussion, would suggest changing much of the language to say “may” – as in, “the negative emotional state may activate caudal…” as the author has not conclusively demonstrated this. Furthermore, the hypotheses that the author proposes testing in order to evaluate the model are not articulated. In general, the Discussion needs editing for clarity of language.

Response: As suggested, “may” has been inserted in several places. Throughout this section, there is now more clarity on specific proposed research directions.

“Behavioral research should be specifically directed toward fear extinction to food and eating.”

“Thus, transcranial magnetic stimulation should be investigated systematically for its role in alleviating cognitive rigidity to alleviate AN related fears.”

“Another biological research direction would be targeting the dopamine system and normalizing hyperresponsiveness to normalize dread response.”

  1. Small note – the methods says “we searched” but there is only one author on the paper. This is disconcerting.

Response: The stated author is the sole contributor to this manuscript and no other individuals have been involved in writing this manuscript, except for staff support for technical submission and proof reading.

  1. Small note – there are places where the references don’t seem to apply to the point being made.

Response: Please be more specific regarding this concern.

Specific comments on the Intro:

  1. The author states that “most deaths” from AN “occur between 16 and 29 years.” That does not seem to match the papers cited. One of them does indicate a higher SMR in that age range, but a different one indicates that deaths increase with duration of illness, and are higher in later decades. 

Response: That part has been removed as it is less relevant to the article and to make space for other text.

  1. The Intro should be more clear that the research from Castro et al is referring to animal research. It would be useful to know how much of this work has been translated to human basic science research.

Response: The introduction now clarifies that the Castro model is based on animal research. It has not been translated to human studies or anorexia nervosa specifically. This has also been added: “This model has not been studied in humans or AN specifically.”

  1. I found the first half of the Intro off-topic from what the paper actually is. Most of the paper is actually a review of findings from animal models, and these serve as the basis for the author’s proposed model, if I understood correctly. If so, Im not sure the description of habit formation is relevant. The author seems to be quickly describing several different models of learning and the result is not that clear. Since the habit formation literature isn’t really part of the science that gets reviewed, it may not be useful in the Intro.

Response: The Castro model integrates higher order motivation with subcortical and hypothalamic circuit response, resulting in approach or avoidance. There are only a few models that have integrated behavior and neurobiology in AN. The reason to describe the so far developed models was to provide a background to support discussion where the models overlap or differ and how Castro’s model can be aligned with those models. I fear that only providing my preferred model and not describing the models that others have developed would counter my goal of bringing the field together toward an integrative approach. To address the concern, I did restructure the introduction though, took away emphasis from the habit model in a separate paragraph, and discuss the models and underlying pathophysiology in one more cohesive paragraph.

  1. Figure 1 is not really explained, and therefore doesn’t contribute to the paper.

Response: The figure has been removed.

Specific comments on the Abstract:

  1. The abstract strings together many jargon terms, and this ultimately obscures the meaning. In addition, some terms are used loosely in ways that seem potentially inaccurate. For example, “food intake control circuitry” is not really established in humans; “central neural response” and “environmental ambience” are unclear phrases. It is not clear in the abstract what the “basic science model” is that the author is proposing.

Response: The abstract has been revised as follows: “Anorexia nervosa is a severe psychiatric illness associated with food avoidance. Animal models from Berridge et al. recently showed that environmental ambience, pleasant or fear inducing, can trigger either appetitive (desire) or avoidance (dread) behaviors in animals via frontal cortex, nucleus accumbens dopamine D1 and D2 receptors, and hypothalamus. Those mechanisms could be relevant for understanding anorexia nervosa. However, models that translate animal research to explain the psychopathology of anorexia nervosa are sparse. This state-of-the-art review surveyed animal and human research of the past ten years to find evidence for whether this model can explain food avoidance behaviors in anorexia nervosa. Research on anorexia nervosa suggest fear conditioning to food, activation of the cortico-striatal brain circuitry, sensitization of ventral striatal dopamine response, and alterations in hypothalamic function. The results support the applicability of the animal neurocircuitry derived model and provides directions to further study the pathophysiology that underlies anorexia nervosa.”

References

Chowdhury, T. G., Chen, Y. W., & Aoki, C. (2015). Using the Activity-based Anorexia Rodent Model to Study the Neurobiological Basis of Anorexia Nervosa. J Vis Exp(105), e52927. doi:10.3791/52927

Grant, M. J., & Booth, A. (2009). A typology of reviews: an analysis of 14 review types and associated methodologies. Health Info Libr J, 26(2), 91-108. doi:10.1111/j.1471-1842.2009.00848.x

Jikomes, N., Ramesh, R. N., Mandelblat-Cerf, Y., & Andermann, M. L. (2016). Preemptive Stimulation of AgRP Neurons in Fed Mice Enables Conditioned Food Seeking under Threat. Current Biology, 26(18), 2500-2507. doi:10.1016/j.cub.2016.07.019

Madra, M., & Zeltser, L. M. (2016). BDNF-Val66Met variant and adolescent stress interact to promote susceptibility to anorexic behavior in mice. Transl Psychiatry, 6, e776. doi:10.1038/tp.2016.35

Monteleone, P., Castaldo, E., & Maj, M. (2008). Neuroendocrine dysregulation of food intake in eating disorders. Regulatory Peptides, 149(1-3), 39-50. doi:10.1016/j.regpep.2007.10.007

O'Connor, E. C., Kremer, Y., Lefort, S., Harada, M., Pascoli, V., Rohner, C., & Luscher, C. (2015). Accumbal D1R Neurons Projecting to Lateral Hypothalamus Authorize Feeding. Neuron, 88(3), 553-564. doi:10.1016/j.neuron.2015.09.038

Round 2

Reviewer 2 Report

This revision did not fully address the comments from the first review. The author made some wording changes, and added a few important additions, but did not substantially change the review, so most of my comments remain the same. Below, I try to offer more specific suggestions for revision. The overall concern is that the writing is too sweeping. It seems that the author has a fairly specific set of ideas about dopamine function and the hypothalamus, which are articulated reasonably at the end of the manuscript. The rest of the literature is not as carefully presented or reviewed, leading to overreach in the claims that are made and an unbalanced presentation. With substantial revision and a sharper focus, this would be a valuable “position paper.”

The major change seems to have been changing the words “systematic review” to “state-of-the-art review.” This remains problematic as it suggests that the author has comprehensively reviewed something. I continue to be concerned that this review presents a very slanted version of the literature that aims only to incorporate the papers that he feels are consistent with the set of data from one body of animal research. The description would be “review” at best. Really, it is mostly a position paper. The Methods section should be removed as it is disingenuous to describe this as a review of the literature when anything deemed “not relevant” for undisclosed reasons was not included. The notion of a hypotheses at the end of the Introduction should also be removed. The authors assertion, in the response to reviewers, that this paper is an attempt to “bring the field together toward an integrative approach” seems disingenuous for a paper that is explicitly trying to explore whether one set of findings in animals can be informative for anorexia nervosa (and has explicitly ignored or given short-shrift to findings that are not consistent).

Overall, I continue to find the review difficult to follow. Some of this may be because in an effort to combine a number of different ideas, the constructs are not fully explained, or contain some imprecision. For example, in the Introduction, in the description of reward-based learning. The dichotomy between habit and goals seems like an over-simplification that can be useful at times, but isn’t so useful here. The habit-based model of AN is not clearly described, nor is it clear how it will be considered within this review. The description included from Dayan and Berridge is fine, but doesn’t explain how this relates to AN. What does this have to do with the EPSI findings in the prior sentence? The author then presents his model of AN from a prior paper. Is the suggestion that this model is related to the habit and goal-directed ideas of the prior two sentences? How would they fit together? This is not explained clearly.

 If the author wanted to clarify this section, there are other bodies of literature that have tried to connect the clinical literature with the neuroscience literature. However, I think that the Introduction might be better spent clarifying what the author is trying to do here in this paper. Perhaps more focused on the actual findings in the animal literature, and better explaining how he hopes to translates this to humans, and to test whether these circuits and this neurophysiology contributes to psychopathology in AN.

In the final section of the Introduction, why are the emotions “desire” and “dread” introduced? What is the purpose of introducing them here? Is this supposed to link to a body of literature on desire and dread? If so, this is not explained. Why does the author think it is helpful to introduce these new concepts? This paragraph goes on to say “this model has not been studied in humans” – what is meant by “model” here? Model of what? I doesn’t seem like a model in animals so much as data from animals.

The author responded to the initial review by changing the title of the first section under results, without making changes to clarify this section. The title still does not reflect the information in the section, but more importantly, the entire section is so cursory that it doesn’t make sense and really doesn’t contribute to this review. It begins by repeating, I think, the specific animal research findings about a circuit that is involved in both consumption and avoidant behavior in rodents (which is framed as desire and dread although those obviously could not have been measured). It is not clear what this has to do with the findings in the next paragraph. The section seems to try summarize very disparate findings and approaches with one sentence each, and each makes broad leaps from the data in each.  For example, how are the ghrelin findings appropriate for this section? Why is DMN discussed here? The link to habit formation in the final paragraph does not relate to what came before it or to the sentence that follows.  I think this entire section could be removed from this paper.

In section 3.2, the author has added a description of the ABA model. It would seem useful to mention that when deprived of food (time-limited access), only some animals become hyperactive and then will choose to use the running wheel excessively (even instead of consuming food even when food is presented). The current description does not make clear that this phenomenon only occurs in some animals -which makes it interesting to understand. It also does not include any mention of intake, just of excessive activity. It would be valuable to explain that this is thought to model some elements of AN – but since not all patients exercise excessively, its not necessarily relevant to all patients, etc.

This section is interesting, but several claims are overstated. The author links the ABA model to dopamine function, and the dread, but has not presented data about dopamine function or about behavioral avoidance in this model. It seems a bit misleading to the reader. Similarly, the limited data suggesting a difference in female mice seems relevant to AN, but the part about prolonged dread seems like an overreach.

This paper would be much clearer if one section presented a clearer report of the findings from the relevant animal studies. This would need to be very clear that this is one of numerous pathways that stimulate (or inhibit) intake in animals. And then to describe the clinical phenomena that the author feels can be better understood by testing this particular pathway in patients with AN.

In the Discussion, the description of aripiprazole indicates that the “evidence is still scant” yet all the citations are open series or case reports. This should more transparently state that there are no randomized or controlled data.

Toward the end of the discussion there is a paragraph referencing metabolic abnormalities and genetics that does not seem related to this review. This whole paragraph can be removed.
